# “Zero Dose” Children in the Democratic Republic of the Congo: How Many and Who Are They?

**DOI:** 10.3390/vaccines11050900

**Published:** 2023-04-26

**Authors:** Daniel Katuashi Ishoso, M. Carolina Danovaro-Holliday, Aimé Mwana-Wabene Cikomola, Christophe Luhata Lungayo, Jean-Crispin Mukendi, Dieudonné Mwamba, Christian Ngandu, Eric Mafuta, Paul Samson Lusamba Dikassa, Aimée Lulebo, Deo Manirakiza, Franck-Fortune Mboussou, Moise Désiré Yapi, Gaga Fidele Ngabo, Richard Bahizire Riziki, Cedric Mwanga, John Otomba, Marcellin Mengouo Nimpa

**Affiliations:** 1World Health Organization (WHO) Country Office, Kinshasa, Democratic Republic of the Congo; 2Kinshasa School of Public Health, University of Kinshasa, Kinshasa, Democratic Republic of the Congo; 3Immunization, Analytics and Insights (IAI), Department of Immunization, Vaccines and Biologicals (IVB), World Health Organization (WHO), 1202 Geneva, Switzerland; 4Expanded Program of Immunization, Kinshasa, Democratic Republic of the Congo; 5National Institute of Public Health, Kinshasa, Democratic Republic of the Congo; 6United Nations Children’s Fund (UNICEF) Country Office, Kinshasa, Democratic Republic of the Congo; 7World Health Organization African Regional Office, Brazzaville, Democratic Republic of the Congo; 8Higher Institute of Medical Techniques of Nyangezi, Public Health Section, Sud-Kivu, Democratic Republic of the Congo

**Keywords:** Democratic Republic of the Congo, immunization, vaccination coverage, zero-dose, inequity, determinants, non-vaccination

## Abstract

(1) Background: The Democratic Republic of the Congo (DRC) is one of the countries with the highest number of never vaccinated or “zero-dose” (ZD) children in the world. This study was conducted to examine the proportion of ZD children and associated factors in the DRC. (2) Methods: Child and household data from a provincial-level vaccination coverage survey conducted between November 2021–February 2021 and 2022 were used. ZD was defined as a child aged 12 to 23 months who had not received any dose of pentavalent (diphtheria-tetanus-pertussis-*Haemophilus influenzae* type b (Hib)-Hepatitis B) vaccine (by card or recall). The proportion of ZD children was calculated and associated factors were explored using logistic regression, taking into account the complex sampling approach. (3) Results: The study included 51,054 children. The proportion of ZD children was 19.1% (95%CI: 19.0–19.2%); ZD ranged from 62.4% in Tshopo to 2.4% in Haut Lomami. After adjustment, being ZD was associated with low level of maternal education and having a young mother/guardian (aged ≤ 19 years); religious affiliation (willful failure to disclose religious affiliation as the highest associated factor compared to being Catholic, followed by Muslims, revival/independent church, Kimbanguist, Protestant); proxies for wealth such as not having a telephone or a radio; having to pay for a vaccination card or for another immunization-related service; not being able to name any vaccine-preventable disease. A child’s lack of civil registration was also associated with being ZD. (4) Conclusions: In 2021, one in five children aged 12–23 months in DRC had never been vaccinated. The factors associated with being a ZD child suggest inequalities in vaccination that must be further explored to better target appropriate interventions.

## 1. Introduction

Routine childhood immunization is one of the most important advances in global health and development [1]. Global routine childhood vaccination programs provided protection to 86% of children in 2019, before the COVID-19 pandemic hit, greatly reducing the effects of diseases such as polio, measles, and several others on children, helping them grow up healthy. Vaccination is considered one of the most cost-effective ways to promote global well-being [2] and even development.

Despite all the proven benefits, vaccination coverage has remained low in some settings, and this was worsened through the pandemic. In 2021, for example, nearly 25 million children remained undervaccinated, 6 million more than in 2019 and the highest number since 2009 [2]. In addition, the number of “children zero dose”, defined as those who did not receive any dose of a diphtheria-pertussis-tetanus vaccine as proxy for lack of access to vaccination services [3], increased from 13.6 million in 2019 to 18.2 million in 2021 [2,3]. Many of these children live in countries affected by conflict, in urban slums, or in remote areas that are hard to reach [2,3], but characterizing them in each country remains important. Gavi, the Vaccine Alliance, through its action plan, has expressed the goal of reducing the ZD children by 25% by 2025 and more than 50% by 2030 [4].

The Democratic Republic of the Congo (DRC), located in the heart of Africa, had an estimated population of 98.3 million in 2016 according to the results of the count organized by the health zones [5]. About 70% of this population lives in rural areas and 30% in urban areas [5]. This population is young, 48% are less than 15 years old, 18.9% are less than 5 years old and about 4% are less than one-year-old [5]. The health system comprises three levels (central, intermediate, and peripheral), and vaccination activities are an integral part of the minimum package of activities of health facilities [5]. The DRC has made significant efforts to improve immunization through the implementation of the Mashako Plan, which is an emergency plan to strengthen the expanded immunization program aimed at reviving routine immunization activities to avoid epidemic outbreaks of certain vaccine-preventable diseases (VPDs) by increasing immunization coverage [5]. Yet, vaccination coverage remains way below the 90% global target according to national surveys and WHO/UNICEF estimates [2]. The DRC is one of the countries in Africa, and the world, with the highest number of ZD children, which results in repeated outbreaks of vaccine-preventable diseases such as vaccine-derived polio, measles, and yellow fever [2,6]. 

The present study was conducted to examine the proportion of ZD children in each of the 26 provinces of DRC in 2021, as well as the factors associated with being ZD, using data from a provincial-level survey. This information will help better characterize this population and serve as a benchmark to evaluate progress.

## 2. Materials and Methods

### 2.1. Study Design

This study is an analytical cross-sectional study aimed at estimating the proportion of ZD children in the DRC and associated factors. It used data from a vaccination coverage survey in 511 of the 519 health zones of the 26 provinces of the DRC conducted between December 2021 and February 2022. Seven health zones were excluded due to insecurity linked to the presence of active armed groups or of poachers.

### 2.2. Sampling 

Sampling was representative at the provincial level, and all 26 provinces of the DRC were surveyed. Multistage sampling was use in each province. First stage: simple random sampling of 5 health areas within each health zone. Second stage: simple random sampling of 30% of avenues/villages within each selected health area. Third stage: systematic random sampling of 34 households with at least one eligible child within each selected avenue/village. Figure 1 summarizes the sampling approach down to the number of children surveyed.

In accordance with the objectives, this survey consisted of 3 distinct statistical units: the child aged 12 to 23 months (though, to estimate other indicators, children aged 6–11 months were also included) living in the household, the head of household sheltering a child aged 12 to 23 months, and the mother/guardian of the child aged 12 to 23 months living in the household. Response rate at the household level was 99.7%.

### 2.3. Data Collection

As mentioned earlier, this is a secondary analysis for data collected from a nation-wide, provincial-level survey that collected data through interview, observation, and document review. The interview was conducted with heads of household, mothers/caregivers of children aged 6 to 23 months, and nurses from health centers or their assistants. The observation and document review were of vaccination cards and health facility registries (for children for whom a card was not seen at home) in order to transcribe vaccination dates into the data collection tool. 

All data collected for the survey was encoded on an android tablet by trained study staff using the SurveyCTO application [7]. All data, including GPS coordinates, was transmitted from the surveyors’ tablets to a secure virtual server after data quality checks were conducted by the field supervisor. Vaccination status was ascertained, by hierarchical order, by the observation of data on the vaccination card or records kept at home, the registers at the health care facilities where available, or the use of recall or verbal history if documented vaccination history was not available.

### 2.4. Variables

The outcome of interest was not having received any dose of the pentavalent vaccine. The explanatory variables used were: household urban/rural location; wealth quintiles calculated from household proxies for socio-economic status using Principal Component Analysis (PCA); presence of a telephone or a radio set as a possible means of communication in the household; maternal/caregiver characteristics including relationship with the child, age, marital status, educational level, occupation, religion, the number of children in the household; gender of the child; birth registration of the child with the civil authority; potential financial barriers from the household such as having to pay for a vaccination card or for another immunization-related service; and caregiver knowledge of vaccine-preventable diseases.

### 2.5. Data Analysis

We performed weighted descriptive analyses of household characteristics in the study sample with categorical variables reported as frequencies and percentages and continuous variables summarized using means and standard deviations or medians and inter-quartile ranges, depending on normality of the distribution.

The extrapolation of ZD children in the general population was made based on the target number of the DRC surviving infants estimated at 4,037,161 for 2021.

We conducted a bivariate analysis between ZD and factors using the Rao-Scott chi-square test, as it is adequate for multistage sampling, to compare the proportions according to the socio-demographic, economic, communicational characteristics, and those related to the system when the expected minimum was ≥5. Then, a multivariable logistic regression model was fitted. The automatic selection of variables using the forward type was used with an entry probability of 0.05. We considered it acceptable after verification of the area under the receiver operating characteristic curve (ROC area = 0.6640). We used the Archer–Lemeshow test to assess the goodness-of-fit of the logistic regression model, as the data was data collected using a complex survey design that involved clustering. Measures of association between each variable and ZD were reported as Adjusted Odd Ratio (AOR) along with their 95% Confidence Intervals (95% CI). Before gauging the final model, we checked the collinearity effect among the variables. The final model only included the variables whose effects remained significant after adjustment.

All analyses were conducted using Stata version 17 (StataCorp, College Station, TX, USA). To account for the complex sampling design, the svy command and the weighting taking into account the multistage design were used.

### 2.6. Ethical Considerations

The Kinshasa School of Public Health Ethical committee approved the vaccination coverage survey before data collection (approval number: ESP/CE/175/2021). Authorization was also provided by health and politico-administrative authorities. Before starting the interview, oral informed consent was obtained from the study participants. The research team provided the respondent with information about the nature of the study, its objectives, the risks and benefits incurred, the freedom to participate or not without any prejudice, the confidentiality, and the contact details of the person in charge of the study for subsequent contact if necessary. Confidentiality was respected by anonymizing the dataset.

## 3. Results

### 3.1. Description of the Sample

#### 3.1.1. Characteristics of Mothers/Caregivers of 12–23 Month Old Children and Gender of the Child

Table 1 describes the characteristics of the sample of 51,054 children aged 12–23 months. Mothers/caregivers had a median age of 27 years (interquartile range (IQR) = 22–33) with the youngest at 13 and the oldest at 81 (a non-mother caregiver), were mostly married (52.5%) or in a free union (38.1%), had completed primary (38.0%) or secondary education (45.5%), were professionally occupied (68.8%, with 42.4% being farmers/breeders), had religious beliefs (98.5%), and had other children under their care (99.9%). Over half of the children 12–23 months were male (54%), and almost all (93.9%) mothers/caregivers cited at least one vaccine-preventable disease (VPD).

#### 3.1.2. Household Socio-Economic, Communicational, and System-Related Characteristics

In terms of household socio-economic, communicational, and system-related characteristics, the majority (89.7%) were in the bottom two wealth quintiles, living in rural areas, with unregistered children (69.4%). Over half had a home radio and telephone. Even though most had not paid money either for the vaccination card or for a vaccination-related service, 39.8% did report having to pay for a card and 21.2% reported having to pay some other immunization-related fee (Table 1).

### 3.2. Proportion of ZD Children 12–23 Months in DRC

The percentage of ZD children aged 12–23 months was 19.1% (95% CI: 19.0–19.2%), which would correspond to a target population of between 767,061 and 775,135 surviving infants in the DRC. This proportion varied importantly between provinces in the DRC. The provinces with prevalence of ZD above the national mean included: Tshopo, Maniema, Sankuru, Mongala, Bas Uele, Tshuapa, Maindombe, Kasai Oriental, Sud Ubangi, Kasai, Haut Uele, and Nord Ubangi (Figure 1).

### 3.3. Factors Associated with Zero-Dose Vaccine in Children Aged 12 to 23 Months in the DRC

After adjusting for independent variables, being zero-dose was significantly associated with the age of the mother or guardian being less than or equal to 19 years (AOR = 1.23 (95% CI 1.06 to 1.44)); maternal education (lack of education AOR = 3.46 (95% CI 1.99 to 5.99), primary AOR = 3.14 (95% CI 1.81 to 5.42), and secondary level AOR = 3.87 (95% CI 2.22 to 6.75) compared to the level of higher or university education); religious affiliation (willful failure to disclose religious affiliation AOR = 4.22 (95% CI 1.63 to 10.94), Muslim AOR = 1.71 (95% CI 1.25 to 2.33), revival/independent Church AOR = 1.35 (95% CI 1.22 to 1.50), Kimbanguist AOR = 1.31 (95% CI 1.03 to 1.66), and Protestant AOR = 1.12 (95% CI 1.01 to 1.25) compared to the Catholic); proxies for wealth such as not having a telephone to use AOR = 1.59 (95% CI 1.45 to 1.75) or a radio AOR = 1.48 (95% CI 1.34 to 1.63); and lack of civil registration AOR =2.04 (95% CI 1.81 to 2.24). Parents who reported having to pay for a vaccination card or for another immunization-related service were more likely to have a ZD child AOR =2.02 (95% CI 1.81 to 2.24) and AOR =3.22 (95% CI 2.57 to 4.03), respectively. Finally, not being able to name any vaccine-preventable disease (VPD) was also associated with being ZD, AOR = 3.37 (95% CI 2.94 to 3.87). Summary of these findings are in Table 2.

## 4. Discussion

The results from this survey showed 19.1% ZD, representing between 767,061 and 775,135 ZD children in the DRC. This result is similar to the proportion estimated by WHO and UNICEF relating to the national vaccination coverage estimates of 19% for the DRC in 2021 [6]. The fact that almost 1 in 5 children aged 12 to 23 months was ZD in the DRC in 2021 is high in comparison to other low- and low-middle-income countries including those in Africa [2]. There was an increase in the prevalence of zero-dose children in sub-Saharan Africa from 6.8% in 2010 to 14% during the COVID-19 pandemic year of 2021 [2].

Our study found that zero-dose children were significantly associated with several factors. Zero-dose children were positively associated with young mothers, which is similar to findings from several studies. The older the mother gets, the less she may hesitate and the fewer barriers she may face to have the child vaccinated [8,9,10,11]. Our study also found that uneducated mothers and those who had only primary or secondary education were more likely to have a ZD child compared to those with higher or university education. Maternal education has been associated with vaccination in most settings [12,13,14,15,16,17,18,19,20,21,22,23,24]. This may be affected by changes affected by education in attitudes, traditions, and beliefs, and even increased autonomy and control over household resources that would improve health-care seeking [12,13,14,15,16,17,18,19,20]. Zero-dose status was also associated with religious affiliation in the DRC, with those not reporting an affiliation having the highest odds of having an unvaccinated child. A pooled cross-sectional study of individual and national data obtained from Demographic and Health Surveys of 33 sub-Saharan African countries found that the children of Muslims were significantly more likely to be zero-dose than children of Christians (25.2% versus 12.3%) [25]. However, Costa et al., in an analysis of 66 low and middle income countries with standardized national surveys since 2010, found that the relationship between religion and vaccination was not consistent across the world [26]. The latter suggests that various cultural and community-level factors may modulate the relationship between religious affiliation and immunization. Working with religious leaders may be an appropriate solution.

Zero-dose children are significantly related to proxies for wealth such as not having a telephone to use or a radio. These two elements are currently important channels through which messages can pass to reach a large part of the population. This significant link somehow reflects the existence of a dissemination of messages likely to encourage parents to have their children vaccinated. However, information and communication are a major challenge in the viability of an initiative. Its success or failure depends on communication and information [27]. Vaccination services, which are the subject of so much controversy, cannot do without communication. Communicating to convince cannot be improvised either at the risk of reaping the opposite effects of what is expected. It is, therefore, worthwhile to rely on the socio-cultural realities of the populations in order to develop appropriate communication strategies, including those adapted to these two channels, to better explain the advantages of vaccination. In the DRC, ZD children were also significantly linked to the lack of civil registration. This result opens a window of action to linking immunization and birth registration, as has been discussed by many in recent years [28,29]. Another finding that is of significance is the high proportion of people reporting having to pay for a vaccination card or another immunization-related fee, as this is a potentially modifiable factor. This study suggests an inhibiting role of fees on child vaccination, and this has been reported as an important barrier elsewhere. This undoubtedly goes against the official free vaccination policy of the DRC, which is aimed at breaking down the financial barrier to give the population maximum access to vaccination services; several studies support that making vaccination free plays a most fundamental role in improving immunization coverage [30,31].

Finally, not being able to name any VPD was also associated with being zero-dose. This association has also been found in several studies conducted in sub-Saharan Africa, particularly in Ethiopia, Burkina-Faso, or Nigeria, including systematic reviews. The lower the level of knowledge, the less likely the caregiver is to vaccinate the child. Working on improving maternal and community knowledge about vaccination, about the diseases that are targeted for protection, the consequences of not vaccinating the child, the vaccination schedule, and on awareness of vaccination campaigns can help improve vaccination [8,17,21,24,32].

The literature suggests that to reduce inequalities in immunization, targeted and pro-equity interventions should be explicitly developed. Such interventions need to be multicomponent to mainly facilitate access through the proper offer of services and community-based mobilization, outreach, and education, adapted to the language and health literacy of the population [33]. Using the results of the 2021–2022 Vaccination Coverage Survey, along with periodic monitoring of process indicators, each province, and health zone, in the DRC is tailoring its immunization delivery strategy. The survey and this analysis were conducted in the context of the Mashako Plan [5]. Alongside other system-strengthening actions, the Mashako Plan is a multipartner and multicomponent initiative that is addressing access and inequalities through simple and targeted interventions developed in collaboration with many stakeholders. The Plan started targeting 9 provinces and has now been extended to all but two of the provinces. It took lessons from previous experiences, including work to improve coverage in Kinshasa [34]. The focus is to favor access to vaccination by strengthening local-level data use and accountability for better micro-planning, outreach, and reduction of vaccine stock-outs, supportive supervision and outreach monitoring, as well as demand generation through community engagement [5].

### Limitations

This study reflects one point in time, and it does not provide longitudinal data. It relies on survey data that can be affected by selection and information biases. The sampling frame was derived from 1984 census data that is known to be inaccurate. To tackle this issue, a household listing exercise was conducted in all selected clusters. Only 7 of 519 health zones were excluded due to insecurity and non-response was 0.3%, with 86920 HHs participating out of 87166 selected HHs. Yet, communities not included in the sampling frame may have been left out and such communities may also be less likely to be reached with vaccines. Vaccination history obtained from cards or facility records may have errors, as records can be incomplete or difficult to read or interpret [35,36]. Additionally, the proportion of vaccination status ascertained by recall was 30%, which can lead to recall bias; although, stating that a child was not vaccinated might be more accurate than indicating which vaccines or how many doses a child has received [37]. Similarly, the ascertainment of factors that relate to vaccination might also suffer from desirability or other biases that are difficult to quantify. Finally, while we assessed factors that were related to being ZD, our study did not go into root cause analysis of the actual reasons for not being vaccinated, or even the factors related to the provision of vaccination services that may affect vaccination.

## 5. Conclusions

Zero-dose is frequent and contributes to the serious health problems in the Democratic Republic of the Congo, with some provinces having over half of their children unvaccinated. Important geographic, demographic, and socio-economic inequalities were observed and quantified. Several factors were associated with not being vaccinated; yet, only better understanding of the underlying causes of ZD will help to inform strategic and operational decisions and to tailor interventions aiming at reducing the ZD burden. Inequalities in immunization should continue to be monitored to assess progress.

## Data Availability

The data presented in this study are available on request from the WHO-DRC office at the email address “nimpamengouom@who.int”. The data are not publicly available due to the sensitivity of certain information from health facilities.

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
