# Peer review of "“Zero Dose” Children in the Democratic Republic of the Congo: How Many and Who Are They?"

_vaccines, 2023, doi:10.3390/vaccines11050900_

Round 1
Reviewer 1 Report
This topic is solidly within the aims of the journal. The study is well done. The areas of improvement are some gaps in the story and some redundancy in the presentation.
The abstract needs to be 2/3 the length that it currently is. The way it is presented is to tell the whole in the abstract, which is not a great strategy.
The transition from the introduction to the materials and method is ackward and incomplete. Authors need to summarize the plan of the project.
The voice and tense of the manuscript is all over the place. The authors should consider sticking with one voice and tense. The manuscript is disjointed like multiple authors wrote it.
The discussion is rehashed version of the results and so the story gets lost in the repetition. I would not repeat the ideas and cut the discussion in half.
The conclusion would be better served with a future studies section and a succinct and precise paragraph that summarizes the story of the paper.
Again, overal a solid submission that needs rework and will be publisheable.
Author Response
Reviewer 1
This topic is solidly within the aims of the journal. The study is well done. The areas of improvement are some gaps in the story and some redundancy in the presentation.
- We thank the reviewer for his/her comments
The abstract needs to be 2/3 the length that it currently is. The way it is presented is to tell the whole in the abstract, which is not a great strategy.
- We have revised the abstract to make it more concise
The transition from the introduction to the materials and method is ackward and incomplete. Authors need to summarize the plan of the project.
- We have worked in improving this transition
The voice and tense of the manuscript is all over the place. The authors should consider sticking with one voice and tense. The manuscript is disjointed like multiple authors wrote it.
- We have tried to revise the grammatical structure with support from a native English-speaker
The discussion is rehashed version of the results and so the story gets lost in the repetition. I would not repeat the ideas and cut the discussion in half. The conclusion would be better served with a future studies section and a succinct and precise paragraph that summarizes the story of the paper.
- We have revised the discussion to make it more concise and also following advice from the other reviewers.
Reviewer 2 Report
Thank you for sharing this article with me. This is an interesting and important paper. I have several minor comments.
1. What was a response rate of the survey?
2. I see some blanks in multivariate table. Why?
3. I'd recommend adding more policy suggestions.
4. Would you discuss limitations such as a lack of longitudinal data and possible selection bias etc.?
Author Response
Reviewer 2
Thank you for sharing this article with me. This is an interesting and important paper. I have several minor comments.
- Thank you for the comments provided.
- What was a response rate of the survey?
- It was 99.7% for the 511 out of 529 health zones included. We have added this information to the methods section.
- I see some blanks in multivariate table. Why?
- Only factors that have been highly statistical significance were included in the model and are shown in the table.
- I'd recommend adding more policy suggestions.
- Thanks for this suggestion. We have reworked the discussion.
- Would you discuss limitations such as a lack of longitudinal data and possible selection bias etc.?
- We have added text to the limitations section
Reviewer 3 Report
This engaging article offers a thorough exploration of the non-vaccination issue in the Democratic Republic of Congo. With a robust methodological design and skillful execution, it requires only minor revisions to enhance its overall quality.
1) The authors should indicate that they use a multistage sampling technique, before explaining in detail the sampling method. For your orientation I e
The sampling method described in line 93 to 100 is a multistage sampling technique, specifically a three-stage cluster sampling design. This approach involves selecting samples at different levels or stages, with each subsequent stage nested within the previous one. Here's a breakdown of the stages
First stage: Simple random sampling of 5 health areas within each health zone.
Second stage: Simple random sampling of 30% of avenues/villages within each selected health area.
Third stage: Systematic random sampling of 34 households with at least one eligible child within each selected avenue/village.)
2) Line 106 “The observation related to the vaccination card in order to note the dates on which the vaccines were received”. is difficult to understand. This appears to be a sentence fragment. Consider rewriting it as a complete sentence. A complete sentence requires a subject and a verb
3) In material and methods.The authors indicate that they performed a weighting, they should be more specific, indicating that they performed a weighting taking into account the multistage design.
4) In material and methods when writing about the Rao-Scott chi-square test explain that was used because is more adecuate for multistage sampling.
5) The more popular name of the Kellie Archer’s test is Archer-Lemeshow test, so is better if the authors write that they used the Kellie Archer’s test or Archer-Lemeshow Test. Also mention that It is used to assess the goodness-of-fit of logistic regression models when the data is collected using complex survey designs that involve clustering.
6) In line 153 indicate if the consent was oral or written
7) The first time the abbreviation IQR (line 164) is used in text, its meaning should be explained.
8) Revie Table 1 Use punctuation marks consistently, sometimes decimals are separated by commas and sometimes by dots.
9) In the discussion section, the authors explore the reasons for the low percentage of non-vaccination among Catholics and associate it with the fact that the Catholic Church has many health institutions in the Congo. Although this could be true, it would be interesting to look for the reasons in the future. The authors could indicate that future research on non-vaccination should include qualitative research with interviews or focus groups in the various religious groups in order to determine the different factors among religious groups that influence non-vaccination.
Author Response
Reviewer 3
This engaging article offers a thorough exploration of the non-vaccination issue in the Democratic Republic of Congo. With a robust methodological design and skillful execution, it requires only minor revisions to enhance its overall quality.
1) The authors should indicate that they use a multistage sampling technique, before explaining in detail the sampling method. For your orientation I e The sampling method described in line 93 to 100 is a multistage sampling technique, specifically a three-stage cluster sampling design. This approach involves selecting samples at different levels or stages, with each subsequent stage nested within the previous one. Here's a breakdown of the stages
- First stage: Simple random sampling of 5 health areas within each health zone.
- Second stage: Simple random sampling of 30% of avenues/villages within each selected health area.
- Third stage: Systematic random sampling of 34 households with at least one eligible child within each selected avenue/village.)
- We thank the reviewer for the clear suggestion. We have revised the text in the Materials and Methods section as per the suggestion, and moved some text that was in the original version to scheme 1.
2) Line 106 “The observation related to the vaccination card in order to note the dates on which the vaccines were received”. is difficult to understand. This appears to be a sentence fragment. Consider rewriting it as a complete sentence. A complete sentence requires a subject and a verb
- We reworded the problematic sentence.
3) In material and methods. The authors indicate that they performed a weighting, they should be more specific, indicating that they performed a weighting taking into account the multistage design.
- We have revised the text in the methods section.
4) In material and methods when writing about the Rao-Scott chi-square test explain that was used because is more adequate for multistage sampling.
- We have added this clarification
5) The more popular name of the Kellie Archer’s test is Archer-Lemeshow test, so is better if the authors write that they used the Kellie Archer’s test or Archer-Lemeshow Test. Also mention that It is used to assess the goodness-of-fit of logistic regression models when the data is collected using complex survey designs that involve clustering.
- We have revised the text in the methods section to provide this clarity
6) In line 153 indicate if the consent was oral or written
- It was oral, we have added this to the text
7) The first time the abbreviation IQR (line 164) is used in text, its meaning should be explained.
- We have spelled-out IQR at first use
8) Revise Table 1 Use punctuation marks consistently, sometimes decimals are separated by commas and sometimes by dots.
- Thank you for pointing out this oversight, we have corrected to use periods as separators for decimals
9) In the discussion section, the authors explore the reasons for the low percentage of non-vaccination among Catholics and associate it with the fact that the Catholic Church has many health institutions in the Congo. Although this could be true, it would be interesting to look for the reasons in the future. The authors could indicate that future research on non-vaccination should include qualitative research with interviews or focus groups in the various religious groups in order to determine the different factors among religious groups that influence non-vaccination.
- We have reworked the discussion section as per the feedback from the reviewers, and provided more suggestions of programmatic interventions, linking it back to the context of the Mashako plan.